# Structural Pharmacology of Cation-Chloride Cotransporters

**DOI:** 10.3390/membranes12121206

**Published:** 2022-11-29

**Authors:** Yongxiang Zhao, Erhu Cao

**Affiliations:** Department of Biochemistry, University of Utah School of Medicine, Salt Lake City, UT 84102, USA

**Keywords:** cation-chloride cotransporters, diuretics, alternate access model

## Abstract

Loop and thiazide diuretics have been cornerstones of clinical management of hypertension and fluid overload conditions for more than five decades. The hunt for their molecular targets led to the discovery of cation-chloride cotransporters (CCCs) that catalyze electroneutral movement of Cl^−^ together with Na^+^ and/or K^+^. CCCs consist of two 1 Na^+^-1 K^+^-2 Cl^−^ (NKCC1-2), one 1 Na^+^-1 Cl^−^ (NCC), and four 1 K^+^-1 Cl^−^ (KCC1-4) transporters in human. CCCs are fundamental in trans-epithelia ion secretion and absorption, homeostasis of intracellular Cl^−^ concentration and cell volume, and regulation of neuronal excitability. Malfunction of NKCC2 and NCC leads to abnormal salt and water retention in the kidney and, consequently, imbalance in electrolytes and blood pressure. Mutations in KCC2 and KCC3 are associated with brain disorders due to impairments in regulation of excitability and possibly cell volume of neurons. A recent surge of structures of CCCs have defined their dimeric architecture, their ion binding sites, their conformational changes associated with ion translocation, and the mechanisms of action of loop diuretics and small molecule inhibitors. These breakthroughs now set the stage to expand CCC pharmacology beyond loop and thiazide diuretics, developing the next generation of diuretics with improved potency and specificity. Beyond drugging renal-specific CCCs, brain-penetrable therapeutics are sorely needed to target CCCs in the nervous system for the treatment of neurological disorders and psychiatric conditions.

## 1. Introduction

Diuretics act primarily on the kidney to increase salt and water excretion, and because they promote diuresis, they are cornerstones of clinical management of fluid overload conditions caused by heart failure, nephrotic syndrome, and liver cirrhosis [1]. Many diuretics, such as mercury, caffeine, digitalis, and acidifying salts, were serendipitously discovered to increase urine production during the 16th to early 20th centuries when kidney functions and its roles in water and salt handling were poorly understood [2]. Modern loop- and thiazide-type diuretics can be traced back to the same class of parental compounds called sulfonamides (Figure 1A) [3]. In the middle of the 20th century, clinicians observed that the sulfonamide class of antibiotics exhibit diuretic activity [4], which incentivized the pharmaceutic industry to pursue sulfonamide derivatives as a new classes of diuretics. These efforts culminated in the development of three novel classes of diuretics between the 1950s and 1970s: carbonic anhydrase inhibitors, loop diuretics, and thiazide diuretics (Figure 1A) [5,6,7]. More than five decades later, these diuretics remain a mainstay of treatment regimens for edema and hypertension, a testament to their transformative impacts in clinical medicine [8]. This remarkable clinical success could easily overshadow an equally important role of diuretics in basic research, in that they have also served as powerful pharmacological tools to dissect renal functions and to discover fundamental ion transport systems. Indeed, shortly after their discovery, loop and thiazide diuretics were soon recognized to antagonize two distinct ion transport mechanisms in the kidney to reduce renal water and salt retention: a Na^+^/K^+^/Cl^−^ transport system functioning at the thick ascending limb of the Henle’s loop and a Na^+^/Cl^−^ transport system operating at the distal convoluted tubule of a nephron [9,10].

The hunt for the molecular targets of loop and thiazide diuretics led to the discovery of cation-chloride cotransporters (CCCs) in the 1990s [11,12,13,14,15,16,17], an exciting era when many other ion channels, transporters, and G-protein-coupled receptors (GPCRs) were also identified as molecular sites of action of well-characterized pharmacological agents or as genetic loci mutated in inheritable human diseases. CCCs belong to the solute carrier 12 family, encoded by *SLC12A1-7* genes. CCCs catalyze electroneutral symport of Cl^−^ together with Na^+^ and/or K^+^ across cellular membranes [18], playing pivotal roles in cell volume regulation, trans-epithelial salt resorption and secretion, and intracellular Cl^−^ homeostasis [19]. CCCs can be grouped into two clades based on their requirement for Na^+^ and/or K^+^ for cotransport of Cl^−^. Na^+^-dependent CCCs consist of NKCC1, NKCC2, and NCC, which are encoded by *SLC12A2*, *SLC12A1*, and *SLC12A3*, respectively. In most cell types, NKCCs and NCC import Cl^−^ using an inward-directed Na^+^ gradient. NKCCs imports Na^+^, K^+^, and Cl^−^ in a 1:1:2 stoichiometry, while NCC mediates influx of Na^+^ and Cl^−^ in a 1:1 stoichiometry. Na^+^-independent CCCs comprise KCC1-4 encoded by *SLC12A4-7* and extrude K^+^ and Cl^−^ in a 1:1 stoichiometry driven by outward-directed K^+^ gradient. The *SLC12A* family additionally includes two less-studied *SLC12A8* (or CCC9) and *SLC12A9* (or CIP1) transporters [20]. Although *SLC12A8* and *SLC12A9* are evolutionarily conserved from worms to flies, fish, and humans, we do not yet fully know what substrates they may transport and what physiological roles they could play at cellular and organismal levels. CCC9 was initially cloned as an amino acid and polyamine transporter [21] and more recently was proposed to function as a nicotinamide mononucleotide (NMN) transporter [22], although this is debatable [23]. CIP1 was first identified to attenuate NKCC transport activity [24] and was later shown to enhance KCC2-mediated ion extrusion [25].

Most CCCs are ubiquitously distributed in tissues and organs in humans, except for NKCC2 and NCC, which are restricted to the kidney, and KCC2, which is predominantly expressed in the nervous system [26]. NKCC2 and NCC specifically localize at the luminal side of renal epithelia cells, where they reuptake salts and obligatory water from the forming urine, contributing to balancing electrolytes and blood volume and pressure (Figure 1B) [27,28]. Indeed, loss-of-function mutations of *SLC12A1* and *SLC12A3* cause salt-wasting and hypotensive Bartter syndrome and Gitelman syndrome, respectively [29,30]. Excessive NCC- and NKCC2-mediated salt reabsorption in the kidney causes hypertension [31], as occurs in Gordon syndrome patients [32]. These patients carry gain-of-function mutations in with-no-lysine kinases (WNKs) or loss-of-function mutations in their E3 ubiquitin degrader KLHL3/CUL3, which promote phosphorylation and activation of NKCC2 and NCC [33,34]. Given the pivotal roles of NKCC2 and NCC in salt handling, loop and thiazide diuretics, which antagonize NKCC2 and NCC, respectively, are widely prescribed for the treatment of hypertension and edema [35]. 

In neurons, KCC2 and NKCC1 function as the major Cl^−^ extruder and accumulator, respectively, and their opposing actions move intracellular Cl^−^ concentration ([Cl^−^]_i_) away from electrochemical equilibrium, maintaining either an inward- or outward-directed [Cl^−^] electrochemical gradient that is dissipated by Cl^−^ channels during synaptic transmission (Figure 1C) [36,37]. KCC2 activity has been shown to progressively increase in the postnatal brain, contributing to maintenance of low [Cl^−^]_i_ in mature neurons as opposed to high [Cl^−^]_i_ in immature neurons [37,38,39]; in humans, KCC2 expression begins at the fetal stage, but its activity is possibly enhanced by reduced inhibitory phosphorylation of two key phosphoacceptor sites [40] and by reduced calpain-mediated degradation in the postnatal brain [41]. NKCC1 is detectable during all stages of brain development [41]. Inhibitory neurotransmitters (e.g., **γ**-aminobutyric acid; GABA) generally stimulate Cl^−^ influx in mature neurons (where [Cl^−^]_i_ is low) but efflux in immature neurons (where [Cl^−^]_i_ is high) via pentameric ligand-gated Cl^−^ channels, resulting in hyperpolarization or depolarization of neurons [42]. Mutations in *SLC12A5* or *SLC12A6*, which encode KCC2 and KCC3, respectively, can cause brain disorders such as epilepsy, seizures, and sensorimotor neuropathy with agenesis of corpus callosum [43,44,45]. These human genetics findings reinforce the concept that brain disorders often share the root cause of an imbalance in excitatory vs. inhibitory neurotransmission. Given the pivotal roles of KCCs and NKCC1 in determining the polarity of response to inhibitory neurotransmitters, they are emerging as attractive targets for developing therapeutic strategies to restore GABA inhibition for the treatment of various brain disorders and psychiatric conditions [46,47]. Bumetanide and other related compounds have been tested in this role in off-label clinical trials, but with debatable outcomes [47,48,49,50]. Bumetanide showed beneficial effects in treating temporal lobe epilepsy, autism, and schizophrenia, but it failed to meet anti-seizure efficacy criteria by itself or as an adjunct with a GABA mimetic in neonatal epilepsy trials. A recent study demonstrated that bumetanide rescues core deficits of a mouse APOE4-related Alzheimer’s disease model and that bumetanide exposure significantly reduces Alzheimer’s disease risk in the population aged 65 and over [51]. This inconsistence reflects etiological heterogeneity of brain disorders but also highlights a major limitation of bumetanide as a repurposed CNS drug, because bumetanide and related compounds bear charged group(s) and thus barely cross the blood–brain barriers (BBBs), resulting in low bioavailability in the brain [52]. 

The pharmacology of CCCs has, for historical reasons, been dominated by loop and thiazide diuretics, with only sparse discoveries of small molecule modulators for NKCC1 and KCCs [53,54,55,56]. The thiazide diuretic hydrochlorothiazide was first approved for the treatment of fluid overload and hypertension in the 1960s [57], shortly followed by furosemide, which was born out of the same parental sulfonamide compounds [58]. The more potent loop diuretic bumetanide was introduced to clinical medicine in 1972 [59]. Other diuretics such as piretanide, torsemide, and ethacrynic acid were developed and approved for clinical use later [60,61,62]. Pharmacokinetics of diuretics partly contributed to their clinical success. For instance, the most prescribed loop diuretic, furosemide, perhaps as well as other diuretics, is actively excreted by the proximal tubule organic transporters into pro-urine [63,64]. Consequently, loop diuretics can flow in tubular fluid to preferentially gain access to apical NKCC2 on distal tubule over systemic NKCC1, despite the fact that they antagonize both NKCC1 and NKCC2 with equal potency in cell-based assays. However, loop diuretics could cause hearing loss by inhibiting NKCC1 in the inner ear in some patients [65]. There are currently no clinical drugs targeting KCC transporters. 

Given that CCCs share high sequence homology and adopt similar three-dimensional (3D) structures with conserved ion binding sites and ion transport mechanisms [66,67], diuretic drugs are limited by their poor specificity, as they often promiscuously inhibit several CCC members [53]. For instance, the loop diuretics, such as bumetanide and furosemide, can inhibit both NKCC1 and NKCC2 with equal potency [68]. Moreover, furosemide can also inhibit KCCs at a higher dose, but whether these off-target actions contribute to its side effects remains unclear [69,70]. Development of next-generation diuretics with improved specificity and potency will benefit enormously from a deep understanding of structures, dynamics, and regulation of CCCs. Atomic-level knowledge of how diuretics or other inhibitors bind to their respective receptor sites on CCCs will provide badly needed structural blueprints for rationally improving their pharmacological properties through medicinal chemistry and virtual screening campaigns. Here, we summarized progresses in structures and pharmacology of CCCs enabled by a recent explosion of cryo-EM structures of CCCs determined in apo state (i.e., ligand free) or bound with diuretic drugs and small molecule inhibitors.

## 2. Structural Pharmacology of CCCs

### Basic Architecture of CCCs

Hydropathy analysis of primary sequences and structural models based on homology to other amino acid-polyamine-organocation (APC) transporter cousins predict that CCCs contain 12 transmembrane helices flanked by large cytoplasmic N- and C-termini [11,12]. The X-ray structure of an isolated C-terminal domain of a bacterial CCC provided the first high-resolution view of a CCC but no insights into ion binding sites embedded in the missing transmembrane domain [71]. Advancement in single-particle cryo-EM has led to a recent explosion of high-resolution structures for NKCC1, NCC, KCC1, KCC2, KCC3, and KCC4 determined in various ionic conditions, in which ion binding sites are completely vacated, partly loaded, or fully loaded (Figure 2) [72,73,74,75,76,77,78,79,80,81,82]. Several CCC structures were determined in complex with diuretic drugs or small molecule inhibitors, providing blueprints for understanding pharmacological regulation of CCCs and for rational development of novel modulators of CCCs [75,78,79,83,84]. 

All CCCs assume a dimeric architecture (Figure 2) [72,73,74,75,76,77,78,79,80,82,83], consistent with previous biochemical studies and fluorescence resonance energy transfer (FRET) measurements [85,86,87]. One notable exception is that KCC4 also intriguingly adopts a monomeric architecture [81], but whether this monomeric structure bears any physiological relevance awaits future investigation (Figure 2). CCC structures all show a well-resolved transmembrane domain (TMD), but some lack density in the large C-terminal domain, possibly because this domain sits beneath the TMD in a range of orientations (Figure 2) [75,78]. When well-defined in cryo-EM maps, two C-terminal domains assemble into a domain-swapped dimer, in which the C-terminal domain of one CCC subunit crosses over the two-fold axis and loosely associates with the TMD of a second subunit [75,78]. The TMD of CCCs adopts a typical LeuT-fold as first observed in a bacterial leucine transporter, in which two inverted repeats of TM1-5 and TM6-10 form the substrate transport core and the remaining two helices (TM11 and TM12) optionally contribute to dimeric assembly [73,75,78]. The TM1 and TM6 helices lie at the center of the ion transport pathway and break 𝛂-helical geometry roughly at the middle of the lipid bilayer, where ion binding sites are organized around these discontinuous hinge regions [73,75,78]. The C-terminal domains of NKCC1, NCC, and KCCs are highly conserved as well, sharing the same ten-stranded β sheet core. However, KCCs and NKCC1/NCC are decorated with distinct extracellular domains located between TM5 and TM6 or between TM7 and TM8, respectively. In KCC2, KCC3, and KCC4 structures, an autoinhibitory N-terminal segment intercalates into and physically plugs the intracellular mouth of the ion permeation pathway, suggesting that this segment must be dislodged from the cytoplasmic vestibule when KCCs actively translocate ions (Figure 2) [77,79,80], for example, upon cell swelling caused by hypotonic challenges, possibly via dephosphorylation of key phosphoacceptor sites. In our recent NKCC1/bumetanide co-structure, a regulatory N-terminal segment bearing phosphoacceptor sites interacts with the C-terminal domain of another subunit (Figure 2), hinting that the strength of this association is likely tuned by the opposing actions of phosphatases and kinases [75].

High-resolution cryo-EM maps of NKCC1, NCC, and KCCs have also unambiguously pinpointed binding sites for Na^+^, K^+^, and Cl^−^ ions (Figure 3A,B), which are further corroborated by molecular dynamic simulations, site-directed mutagenesis, and an ion coordinating geometry analogous to that seen in other ion-coupled transporters [72,76]. In the zebrafish NKCC1 (DrNKCC1) structure, K^+^ is coordinated by the mainchain carbonyl oxygens of N220 and I221 (TM1), P417 and T420 (TM6), and the sidechain oxygen of Y305 (TM3) and T420 (TM6) [72]. K^+^ was later found to be coordinated in the same octahedral geometry by an equivalent set of residues in KCCs [76]. Of note, Y305 is conserved in K^+^-transporting NKCCs and KCCs but is replaced by a histidine residue in NCC [72,76]. This tentatively explains why NCC does not transport K^+^, as it lacks this key tyrosine residue to coordinate K^+^. Instead, this histidine participates in coordination of a Na^+^ ion in NCC [82]. Most NKCC1, KCCs, and NCC maps also resolve two strong non-proteinaceous densities assigned to Cl^−^_i_ and Cl^−^_ii_ based on the local chemical environment [72,76,77,78,79,80]. In the DrNKCC1 structure, the Cl^−^_i_ ion sits within an interaction distance extracellular to the K^+^ ion, coordinated by the mainchain amide nitrogen atoms of G223, V224, and M225 on TM1 [72]. The Cl^−^_ii_ ion observed in the DrNKCC1 map is coordinated by mainchain amide nitrogen atoms of G421, I422, and L423 on TM6, as well as the sidechain oxygen of Y611 on TM10 [72]. It is intriguing that KCCs harbor two Cl^−^ binding sites vs. one K^+^ site, as this stoichiometry appears to contradict the electroneutral cotransport of 1 K^+^: 1 Cl^−^. In one model proposed to rationalize this structural finding, Cl^−^_ii_ may be an allosteric ion that is not transported, remaining bound to the transporter throughout a transport cycle. In this model, Cl^−-^_i_ and K^+^ are transported together, as they are coupled via electrostatic interactions. In another model, there may be a second as-yet unknown K^+^ site in KCCs, in congruence with the electroneutral ion transport using a 2:2 instead of 1:1 stoichiometry. One important corollary of this model is that the second K^+^ site only exists in KCCs, not in NKCCs, so as not to break the electroneutrality of NKCCs. Na^+^ may more loosely bind to NKCC1 than K^+^ and Cl^−^ do, as only weak density can be observed in the putative Na^+^ site of NKCC1 maps [72,75,88]. Notwithstanding this caveat, the Na^+^ site can be inferred by analogy from the conserved Na^+^ coordination geometry of so-called Na^+^ site 2 observed in other Na^+^-coupled APC transporters [89,90,91]. In DrNKCC1, Na^+^ is coordinated by the backbone carbonyl oxygen atoms of L219 and W222 on TM1, A535 on TM8, and the sidechain oxygens of S538 and S539 on the TM8 helix [72]. This speculated Na^+^ site has been validated by mutagenesis studies and molecular dynamic simulations of NKCC1 [72]. This Na^+^ site is conserved in NCC and coordinates a second Na^+^ ion, suggesting that NCC possibly shuttles 2 Na^+^: 2 Cl^−^ for each transport cycle.

## 3. Extracellular Vestibule: A Hotspot for Pharmacological Inhibition of CCCs 

CCC pharmacology, for historical reasons, remains dominated by loop and thiazide diuretics. Recent high-throughput screening for small molecule modulators of KCC2 discovered a potent inhibitor (VU0463271) that antagonizes all KCC isoforms [55]. An activator of KCCs (CLP257) was similarly discovered, and in a mouse spinal cord injury model, this small molecule promotes formation of functional neuronal circuits and hence recovery from paralysis, possibly by stimulating KCC2-mediated Cl^−^ extrusion [56,92]. However, whether this potentiator directly acts on KCCs remains controversial [56,93]. A computation-aided drug design yielded an NKCC1-specific inhibitor (ARN23746) that holds great promise in preclinical studies for the treatment of brain disorders, because, unlike bumetanide, it can cross the BBB to antagonize NKCC1 within the brain [54]. Elucidating the precise binding poses and inhibitory mechanisms of these pharmacological reagents represents a major goal of structural pharmacology of CCCs.

Recent cryo-EM structures of human KCC1 bound with VU0463271, human NKCC1 individually bound with bumetanide and furosemide, and human KCC3 in complex with [(dihydroindenyl)oxy] acetic acid (DIOA) afford the first atomic views of how these inhibitors interact with and affect ion transport of different CCC transporters (Figure 3A–C) [75,78,79,83]. In a human NKCC1/bumetanide co-structure, bumetanide wedges into the extracellular ion permeation path and displaces the Cl^−^ ion from the Cl^−^_i_ site, establishing interactions with Val385 on TM3, Leu671 and Ala675 on TM10, and K^+^ that would otherwise be coupled to Cl^−^_i_ via electrostatic interaction (Figure 3A) [75]. The NKCC1/bumetanide structure rationalizes two key findings from previous kinetics measurements of bumetanide binding to NKCC-expressing tissues or cells [94]. First, all three ions (K^+^, Na^+^, and Cl^−^) are indispensable for bumetanide binding. Second, Cl^−^ shows a peculiar biphasic effect, such that it promotes bumetanide binding until ~5 mM and then progressively inhibits bumetanide binding to NKCC1 with increasing [Cl^−^]. The structure also tentatively explains why different loop diuretics exhibit a wide range of potency and could guide rational design of a new generation of loop diuretics. For instance, the phenoxyl group of bumetanide interacts with Val385 on TM3, as well as Leu671 and Ala675, located on TM10. In furosemide, this phenoxyl group is substituted by a chlorine atom, likely explaining why it is ~ 50-fold less potent in inhibiting NKCCs than bumetanide. The carboxyl group of bumetanide interacts with the K^+^, which may explain why loop diuretics with this carboxyl group are more potent than non-carboxyl loop diuretics such as ethacrynic acid. Bumetanide interacts with a set of residues lining the extracellular ion permeation pathway that are invariant in NKCC1 and NKCC2, explaining why bumetanide (and other loop diuretics) cannot discriminate between these two NKCC isoforms. Medicinal chemists have extensively explored bumetanide derivatives for the discovery of NKCC1-specific inhibitors without success [54,95]. Our NKCC1/bumetanide now suggests that a more enticing strategy to achieve this unmet goal is to develop small molecules/biologics to target their respective regulatory mechanisms that must have adapted to fulfill their distinct roles in the kidney (NKCC2) and brain (NKCC1). A more recent study showed that NKCCs possibly bear an additional loop diuretic binding site at the C-terminal domain [83].

In the KCC1/VU0463271 co-structure, VU0463271 also occludes the extracellular ion entryway by fitting into an extracellular pocket formed in an outward open state by TM 1b, 6a, 3 and 10 helices (Figure 3B) [78]. VU0463271 is an elongated molecule and reaches deeper into the ion translocation pathway than bumetanide. In fact, VU0463271 dislodges K^+^ from its binding site. Nonetheless, both VU0463271 and bumetanide arrest CCCs in an outward open conformation, sterically hindering isomerization to other transport states for completion of a transport cycle. VU0463271 establishes a multitude of polar and hydrophobic interactions with residues along the extracellular ion transport pathway. At the mouth of the extracellular entryway, the 4-methyl-2-thiazolyl group of VU0463271 wedges between Arg140 and Glu 222, breaking the salt bridge that closes the extracellular gate presented in the inward-open structures of KCC1. Here, the 4-methyl-2-thiazolyl group coordinates with the residues Glu222, Ile223, and Tyr227 on TM3, as well as Val135 on TM1b via hydrogen bonding and hydrophobic packing interactions. Deeper toward the central ion binding sites, the phenyl-3-pyridazinyl group engages with a number of residues located on TM6a, TM3, and TM10, including Tyr216, which would otherwise participate in coordination of a permeating K^+^ ion. The fact that VU0463271 and K^+^ share an overlapping binding site provides a plausible explanation of why VU0463271-related compounds are K^+^-competitive inhibitors, with its potency progressively diminished as [K^+^] increases in cell-based ion flux assays [55]. 

In the cryo-EM map of KCC3 bound with DIOA (a non-specific KCCs inhibitor), two DIOA molecules were observed in the central cleft between the two TMDs (Figure 3C) [79]. The indanyl group of DIOA interacts with hydrophobic residues from TM10, as well as the scissor helix and TM12 of the second subunit. The carbonyl group and the carboxyl group of DIOA interact with K664 in TM12 and a conserved R617 residue in the intracellular loop connecting TM10 and TM11. DIOA induces no noticeable conformational changes when compared to KCC3 structure determined in an inhibitor-free (apo) state. DIOA thus stands in contrast to bumetanide, furosemide, and VU0463271, which share the same inhibitory mechanism of arresting CCCs in an outward open state. DIOA was proposed to rigidify TMDs and/or intracellular loop, thus inhibiting KCCs by preventing conformational changes associated with ion translocation.

Neurotransmitter reuptake transporters arguably enjoy the richest pharmacology within the APC family, including therapeutics such as antidepressants and abusive substances such as cocaine [96]. Some of these psychoactive agents directly compete with substrates for a shared binding site along the transport pathway, and they are thus classified as substrate competitive modulators; others allosterically regulate transport rate by stabilizing a specific conformational state, and they are referred to as allosteric modulators (Figure 3D) [96]. Based on this conceptual framework, loop and thiazide diuretics and VU0463271 directly target extracellular ion translocation pathways and inhibit CCCs more or less in a substrate-competitive manner (Figure 3A,B). This explains why all these compounds show poor isoform specificity, as the ion permeation pathways of CCCs are lined by a set of highly conserved residues. 

## 4. Alternate Access Model of CCCs

Most CCC structures have been captured in an inward open state irrespective of ionic conditions intended to maintain CCCs fully or partially loaded with substrate ions [72,73,74,76,77,79,80,81]. This is somewhat intriguing, because inclusion and exclusion of substrates have been successfully used to bias other transporters into a certain transport state [97,98]. We suspect that ionic gradients across a membrane may be required to drive conformational changes as CCCs proceed along their transport cycle. Alternatively, because CCCs are phosphorylated at multiple phosphoacceptor sites within cytosolic domains such that their activity could be regulated in a graded manner according to a phosphorylation code, recombinant CCC proteins used for structural studies so far lack these key post-translational modifications, which could be indispensable for CCCs to be conformationally responsive to ion substrates. 

We recently used loop diuretics (bumetanide and furosemide) and VU0463271 to bias NKCC1 and KCC1 toward an outward-open state that would be otherwise rarely populated in vitro [75,78]. Both NKCC1 and KCC1 undergo similar conformation changes as they isomerize between inward and outward open states (Figure 4A). The TM1, TM2, TM6, and TM7 helix bundle of KCC1 and NKCC1, which is commonly referred as the “core domain” in LeuT fold transporters [99], remains static during the transport cycle (Figure 4A). On the contrary, the TM3, TM5, TM8, and TM10 of CCCs, which is referred to as the “scaffold domain”, undergo subtle rock movements, resulting in reciprocal opening and closing of the extracellular and intracellular ion transport pathways (Figure 4A). Notably, TM1 and TM6 half helices of CCCs do not undergo hinge-bending motions around the discontinuous regions as seen in LeuT and many other APC transporter cousins (Figure 4B) [99,100]. Such subtle conformational changes can accommodate coordination and release of small inorganic ions but not much larger organic molecules (e.g., amino acids) transported by other APC transporters. As subtle conformational changes tend to occur in a shorter time scale, this may explain why CCCs can mediate much higher rates of substrate movement across cell membranes than many other related APC transporters [101]. Of note, we are yet to construct a complete transport cycle of CCCs in structural terms, as we still lack a structure of CCC determined in an occluded state.

## 5. Plasticity in Mode of Assembly of CCC Dimers 

As we and others have demonstrated in KCCs and NKCC1 structures [72,75,76,78,80], CCCs can remarkably assume multiple dimeric architectures via selective formation of inter-subunits interfaces between extracellular domains, transmembrane domains, and cytoplasmic C-terminal domains (Figure 2). For instance, NKCC1 adopts three distinct dimeric isoforms (Figure 5). In the first form, represented by the human NKCC1/bumetanide structure (Figure 5A), two TMDs are separated within the bilayer, while the C-terminal domain dimer tightly associates with the TMDs [75]. In the second form (Figure 5B), the TMDs of two subunits associate via the inverted-V-shaped TM11-turn-TM12 structure, while the C-terminal domains are seen to only loosely associate with the TMDs [72]. In the third form (Figure 5C), TMDs of two subunits are superimposed with the second form, whereas the large C-terminal domains are not well-resolved, possibly because they sit beneath the TMDs in a range of orientations [73]. The C-terminal and transmembrane domain interface is obviously malleable, as it is only seen in the TMD-separated NKCC1 dimeric form [75]. Of note, the conserved extreme C-terminal tail (Arg1201-Ser1212) is a key component of the above interface, interacting with both an N-terminal phosphoregulatory segment and intracellular loop 1 (ICL1), which connects TM2 and TM3, hinting at potential allosteric communications among these structural elements that may be subject to regulation by kinases and phosphatases [75].

Human KCC1 adopts at least two distinct dimeric architectures (Figure 5). One KCC1 dimeric form resembles all other known KCC structures in which an inverted-V-shaped helix–turn–helix structure formed by TM11 and TM12 constitutes the principal dimeric interface within the lipid bilayer in addition to the interdigitating cytosolic C-terminal domains (Figure 5D) [77,78]. In the second form represented by KCC1/VU0463271 structure (Figure 5E), cytosolic domains are not resolved and likely do not contribute to dimeric assembly [78]. Direct protein contacts within the lipid bilayers are also insignificant. Instead, an extracellular domain, formed by a stretch of ~120 residues between TM5 and TM6, participates in homotypic interactions with the same structure from a second KCC1 subunit.

The plasticity in modes of dimeric assembly of NKCC1 and KCC1 raises important questions for future studies: (1) Do all these CCC dimeric forms exist in native cells? (2) Do these distinct CCC dimeric forms exhibit intrinsically different ion transport rates? (3) Could conversion between these dimeric forms be intimately associated with activation of CCCs by (de)phosphorylation or by engagement of cellular factors such as creatine kinase, Neto2, and gephyrin, and if so, how [102,103,104,105]?

## 6. Concluding Remarks

The recent breakthrough in determining structures of CCCs has greatly advanced our knowledge of their dimeric architectures, their ion binding sites, and the mechanisms of action of loop diuretics and small molecule inhibitors. However, several important questions remain outstanding. First, we need to develop CCC pharmacology beyond loop and thiazide diuretics. CCCs are validated targets for the treatment of edema, hypertension, and neurological disorders. One testament to the druggability of CCCs is that loop and thiazide diuretics remain mainstays for the management of hypertension and fluid overload by antagonizing renal NKCC2 and NCC. However, we lack brain-penetrable compounds and biologics to target CCCs in the nervous system for the treatment of brain disorders. Additionally, loop and thiazide diuretics are successful clinical treatments, but they have limitations of poor specificity and low potency. For instance, most diuretics promiscuously inhibit multiple CCC isoforms, as CCCs exhibit high sequence and structural conservation. Recent NKCC1/bumetanide and KCC1/VU0463271 structures provide the first insights into the inhibitory mechanisms of CCCs (Figure 3A and 3B). We hope that these co-structures will spark interest among medicinal chemists in synthesizing derivatives as a new generation of diuretics. Moreover, computational docking of virtual chemical libraries into CCC structures may discover compounds with novel chemical scaffolds distinct from existing modulators of CCCs. Finally, biologics such as peptides, nanobodies, and antibodies could afford exquisite specificity to discriminate closely related CCC isoforms, which is highly desirable but may prove difficult to achieve with small molecule-based therapeutics. 

How CCCs orchestrate conformational changes to translocate ions remains elusive. CCC structures now lay a foundation for molecular dynamic simulations to uncover routes through which ions can gain access to and escape their respective binding sites. CCC structures captured in inward and outward open conformations provide the first snapshots to infer how CCCs isomerize between these two states, but more intermediate conformation structures (e.g., occluded state) are needed to fully understand the transport cycle of CCCs. Finally, the cryo-EM method is well-suited to capturing dominant conformational states of a given protein, and we expect that single-molecule FRET (smFRET) and double electron–electron resonance (DEER) could be more fruitful approaches to study dynamics of CCCs and to pinpoint rate-limiting conformational transitions associated with CCC-mediated ion translocation.

Finally, the structural basis for phosphoregulation of CCCs remains a major mystery. CCCs are regulated downstream of the WNKs-SPAK/OSR1 kinase pathway, which generally activates NKCCs and NCC but inhibits KCCs [106]; phosphatases principally counteract kinase actions, inactivating NKCCs and NCCs while activating KCCs [107].

Although phosphoacceptor sites in CCCs have been identified within their cytoplasmic N- and C-terminal domains [108,109,110], we have only started to appreciate functional consequences of (de)phosphorylation of each site. For instance, in KCC2, posttranslational modifications of some phosphoacceptor sites have negligible effects on its transport activity, while phosphorylation of others sites can either stimulate or attenuate transporter function [111,112]. It is also not clear how (de)phosphorylation of these cytosolic motifs can trigger a wave of conformational changes that propagates to eventually impact ion permeation pathways embedded within the transmembrane domain. In this regard, CCCs intriguingly adopt distinct dimeric architectures, one of which may be conducive for rapid isomerization among transport states and thus exhibit a higher transport rate than other dimeric forms. A corollary of this hypothesis is that (de)phosphorylation of CCCs may regulate ion transport activity via preferentially stabilizing one dimeric form over others (Figure 5). A combination of structural and molecular dynamics studies to directly compare dephosphorylated and authentically phosphorylated CCC samples will hopefully solve the mystery of phosphoregulation of CCCs. An atomic-level understanding of CCC phosphoregulation will facilitate development of isoform-specific CCC modulators for therapeutic applications by rationally targeting their phosphoregulatory apparatuses, which could have evolutionarily adapted to operate in diverse physiological contexts (e.g., Cl^-^ homeostasis in neurons and salt reabsorption in the kidney) as opposed to ion translocation pathways of CCCs that are lined by almost invariant residues. Indeed, key phosphoacceptor sites are mostly enriched in the N-terminal region of NKCCs and NCC and the C-terminal domain of KCCs.

## Figures and Tables

**Figure 1 membranes-12-01206-f001:**
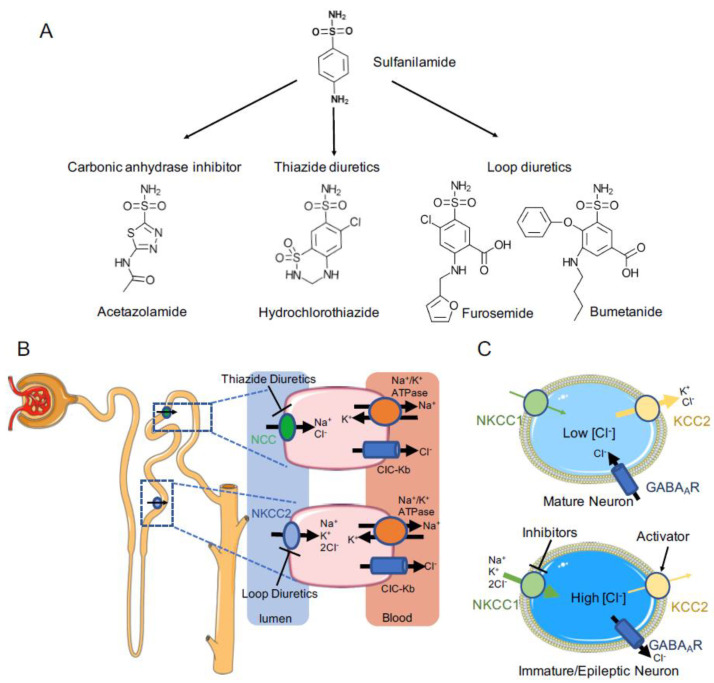
Cation−chloride cotransporters are the molecular targets of diuretic drugs in the kidney and regulate polarity of GABA response in nervous system. (**A**) Sulfanilamide inspired the development of three classes of diuretics. (**B**) Loop and thiazide diuretics inhibit NKCC2− and NCC−mediated salt and water retention in the kidney, respectively. (**C**) NKCC1 and KCC2 function as the major Cl^−^ loader and extruder in neuron, respectively. Their opposing actions are critical in setting [Cl^−^]_i_ in neurons and, consequently, the polarity of GABA response.

**Figure 2 membranes-12-01206-f002:**
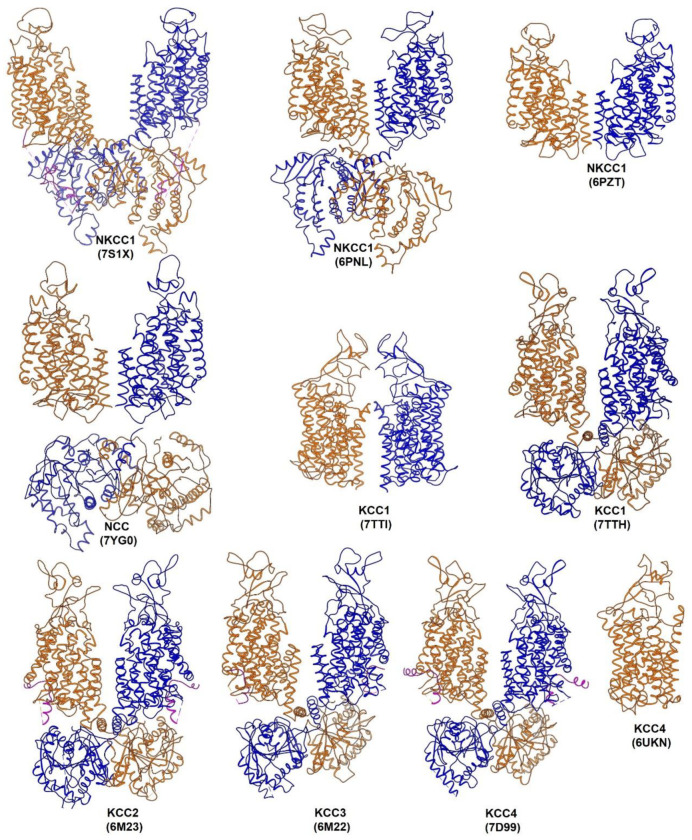
A gallery of cryo−EM structures of CCCs. All CCCs assume a dimeric architecture, except that KCC4 can also exist as a monomer. Two subunits in a CCC dimer are colored red and blue. The NKCC1 N−terminal phosphoregulatory segment and KCC N−terminal autoinhibitory segment are shown in magenta.

**Figure 3 membranes-12-01206-f003:**
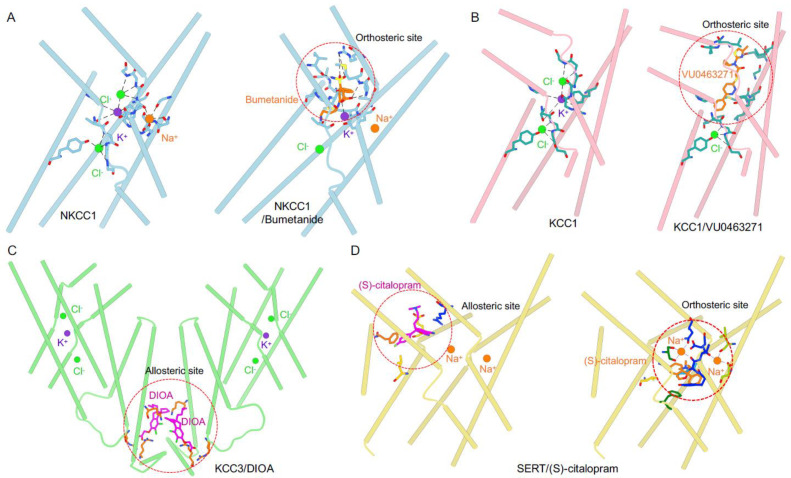
Receptor sites for CCCs and serotonin transporter (SERT) modulators. (**A**,**B**) Bumetanide and VU0463271 act as orthosteric inhibitors of NKCC1 and KCC1, respectively. (**C**) DIOA may be an allosteric inhibitor of KCCs. (**D**) (S)−citalopram binds to both allosteric and orthosteric sites on SERT. Small molecule inhibitors are rendered as sticks; ions are shown as spheres.

**Figure 4 membranes-12-01206-f004:**
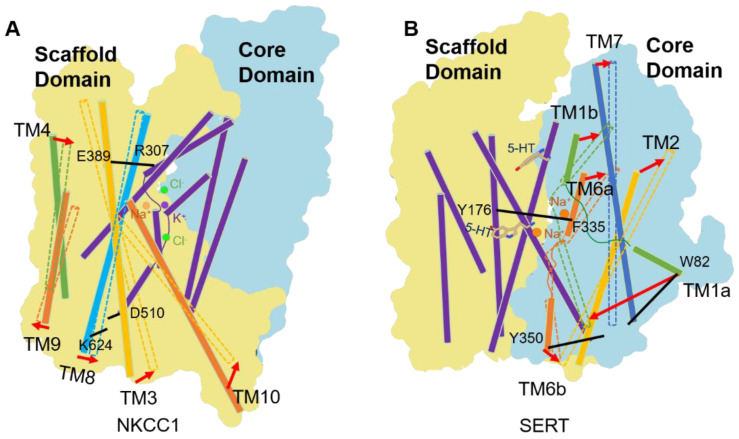
Comparison of alternate access model for (**A**) NKCC1 and (**B**) SERT. NKCC1 and SERT show opposite mobile domain vs. static domain when they isomerize between inward and outward open states. Static helices are shown in purple. Ions are shown as spheres, substrate 5-HT is rendered as sticks. Movements of helices are indicated as arrows. Residues that gate access and escape of substrates are also indicated.

**Figure 5 membranes-12-01206-f005:**
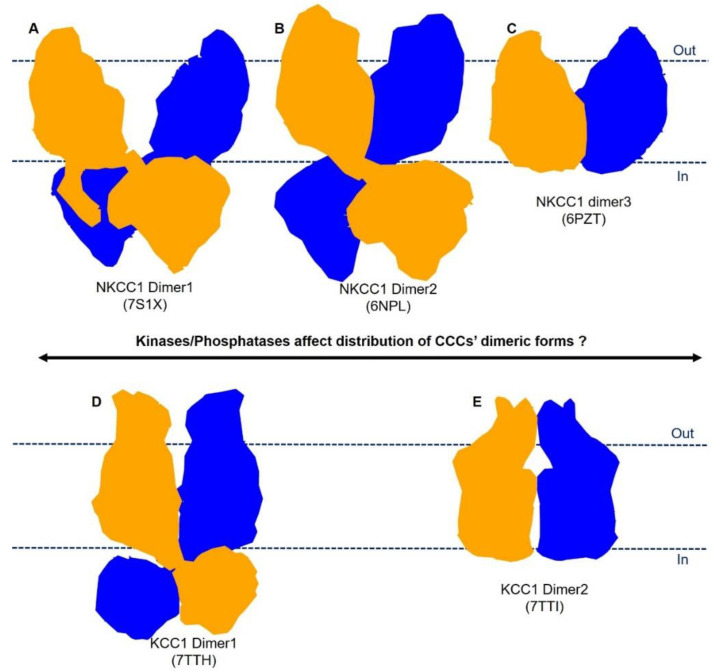
A hypothetical model of phosphoregulation of CCCs. NKCC1 and KCCs assume diverse dimeric architectures, which may have inherently distinct transport rates. Conversion among these dimeric forms may be subject to regulation by the opposing actions of kinases and phosphatases. (**A**–**C**) show the different dimeric forms of NKCC1 and (**D**,**E**) show different dimeric forms of KCC1.

## Data Availability

Not applicable.

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
