# Peer review of "Structural Pharmacology of Cation-Chloride Cotransporters"

_membranes, 2022, doi:10.3390/membranes12121206_

Round 1
Reviewer 1 Report
The review manuscript titled as ‘Structural Pharmacology of Cation-Chloride Cotransporters’ by Zhao and Cao, summarize the current knowledge about structures of cation-chloride cotransportersl and their mechanisms of action of loop diuretics and small molecule inhibitors. Indeed, they provide a comprehensive summary about the current knowledge in this field and end up with a perspective about developing next generation of diuretics with improved potency and specificity, especially developing brain-penetrable therapeutics by targeting CCCs for the treatment of neurological disorders, and many currently unsolved questions in the fields. This work should be of wide interests to most researchers in membrane biology and biomedicines of CCCs.
Only some minor comments,
Figure legends are needed for most of the figures, for examples for Figure 2 “TMD colored in green, CTD in marine and scissor helix in yellow……”.
Reviewer 2 Report
This review provides a satisfying and up-to-date analysis of current three-dimensional structures of CCCs in relation to 1) small molecules binding 2) ion transport and 3) dimeric organization of these transporters. The authors pointed out the need to take advantage of 3D structures to invest on structure-based drug design campaigns. They also critically analyzed possible limitations of the structures due to partial understanding of phosphorylation-dependent conformations, which are critical for CCCs modulation, particularly for those whose activity is dowregulated in diseases (such as KCC2 in brain diseases). I did not find major criticisms to the content of the review. One minor adjustment could be done on Fig. 5 to better visualize the opposing effect of kinases/phophatases activity on CCCs dimeric organization. I felt the arrow was not really explicative.
Reviewer 3 Report
In this review Zhao and Cao describes the structural organizations of NKCC1 and KCCs. They compare the structures due to the whole structure, ionic binding sites, dimeric structure and the alternate access model. A special focus was laid on the binding of inhibitory drugs within the structure. I have made several major points, that we authors please shall address.
Major points:
Introduction:
P2: line 54-73: Please be more precise in mentioning the phylogenetical relationships of CCCs.
P2: line 68-73: Please mention to which protein SLC12A8 and SLC12A9 belong to. The functioning of CCC9 is well characterized (see Daigle et al. 2009). For CIP1 please have a look at these publications: Caron et al. 2000 and Wenz et al. 2009.
P3: line 92-94: Please be more precise in mentioning when KCC2 is present in immature or mature neurons. This dependence on the neuronal cell population and there are differences if the expression of KCC2 increases or if KCC2 is present in immature neurons, but became active during development.
P3: line 94-97: The sentence “Inhibitory neurotransmitter …” is not clear for me. Be more precise when GABA results in hyperpolarization or depolarization dependent on the intracellular Cl- concentration that is mediated by NKCC1 or KCC2. What is happening with NKCC1 during development?
P2-p3: Bumetanide at low dosages affect NKCC1 and NKCC2. Furosemide, as far as I know, targets KCC1-KCC4 and also NKCCs. Please mention this and also discuss due to the broad expression pattern of some of the CCCs which side-effects these drugs have. Be clearer here.
Structural pharmacology of CCCs:
Fig. 2 and also p6 line 154-156: Actually, I doubt that KCC4 is the only CCC that exists as a monomer…. In this structure the C-terminus is missing… Is it not possible that KCC4 naturally exists as a dimer and this could be visualized in a Cryo-EM structure of the whole KCC4 protein?
P6: line 174-175: You mention that the dislocation of the N-terminus will be regulated via cell swelling. Are there any known motifs that are responsible for that? Or is this regulated via (de)phosphorylation?
P6: line 189-192: “However, substitution of this histidine in NCC with tyrosine failed to confer K+ transport activity, indicating a more nuanced ion transport mechanism as an ancestral CCC gene duplicate and then diverges along distinct evolutionary paths to fulfill their diverse physiological roles” I don`t get that point. Please have a look on the phylogenetical relationship of the CCC family and then state the exact phylogenetical step at which NCCs become more different than NKCCs.
Extracellular vestibule: a hotspot for pharmacological inhibition of CCCs:
P7: line 219-222: Why is it important to have a KCC2 inhibitor as a clinical compound? Please provide the clinical background why it is necessary to have activators or inhibitors to interfere NKCC1 and KCC2 function (esp. in epilepsy, ASD or schizophrenia).
P7 line 232: What does DIOA do?
4. Alternate access model of CCCs
Figure 4.: Could you please insert the substrate binding sites into the figure (in CCCs and SERT). How does ion binding in CCCs affect the conformational changes?
P9 line 324-325: “This may also explain why CCCs can mediate much higher rates of 324 substate movement across cell membrane than many other related APC transporters” Could you please explain this in more detail?
5. Plasticity in mode of assembly of CCC dimers
P9-10 line: 333-350: Can you please directly match the three described NKCC1 dimeric structures to the three NKCC1 illustrations depicted in Fig. 5 (for ex. Fig. 5A, 5B, 5C à dimeric structures of NKCC1). And please adopt this to the KCC structures.
P10: line: 370: Why do you emphasize the creatine kinase? What is special about it? And why don`t you mention kinases that are more prominently described for CCCs?
6. Concluding remarks
P11: line 403-407: The regulation of CCCs via (de)phosphorylation, esp. for KCCs is much more complex as it was described here. For example, KCC2 can be activated via dephosphorylation and phosphorylation dependent on the specific phospho-site. Please be here more precise.
P11: line 416-420: Could you please describe the last sentence more deeply.
Minor points:
P2: line 68-73: SLC12A iterative
P5: line 162: leuT-fold à LeuT-fold
P6: line 171: “in some KCC structures” à which?
P6: line 189-192: Substitution of his to tyr in NCCs à Is this the primary reference: 59? I guess this is a review article. Please mention the primary reference.
Round 2
Reviewer 3 Report
Dear Prof. Zhao and Prof. Cao,
I do not have anything else to complain and I therefore accept this article as it is.